# Slightly broken higher-spin current in bosonic and fermionic QED in the large-N limit

Zheng Zhou (周正)[1,2] and Yin-Chen He[1]

**1** Perimeter Institute for Theoretical Physics, Waterloo, Ontario, Canada N2L 2Y5
**2** Department of Physics and Astronomy, University of Waterloo,
Waterloo, Ontario, Canada N2L 3G1

## Abstract

We study the slightly broken higher-spin currents in various CFTs with U(1) gauge field, including the tricritical QED, scalar QED, fermionic QED and QED-Gross-Neveu-Yukawa theory. We calculate their anomalous dimension by making use of the classical non-conservation equation and the equations of motion. We find a logarithmic asymptotic behaviour ($\gamma_s \sim 16/(N\pi^2)\log s$) of the anomalous dimension at large spin $s$, which is different from other interacting CFTs without gauge fields and may indicate certain unique features of gauge theories. We also study slightly broken higher-spin currents of the $SU(N)_1$ WZW model at $d = 2+\epsilon$ dimensions by formulating them as the QED theory, and we again find its anomalous dimension has a logarithmic asymptotic behaviour with respect to spin. This result resolves the mystery regarding the mechanism of breaking higher spin currents of Virasoro symmetry at $d = 2+\epsilon$ dimensions, and may be applicable to other interesting problems such as the $2 + \epsilon$ expansion of Ising CFT.



## 1   Introduction

Conformal field theories (CFT) are a family of quantum field theories with fundamental importance and wide applications in physics, which include quantum gravity as well as critical phenomena in condensed matter physics. One interesting class of CFTs is critical gauge theories in 3$d$, which describes various exotic critical phases [1–3] and phase transitions [4–9] in quantum matter systems. Theoretically, these critical gauge theories are not well understood and pose challenges for modern CFT techniques such as the conformal bootstrap [10].

A wealth of physical properties of a CFT are determined by its operator spectrum labelled by ($\Delta$, $s$): $\Delta$ is the scaling dimension, and $s$ is the Lorentz spin characterising how the operator transforms under the Lorentz rotation symmetry SO($d$). The spin-$s$ ($s \geq 1$) operators $J^{\lambda_1 \cdots \lambda_s}$ of a unitary CFT are known to satisfy the unitary bound $\Delta_s \geq d - 2 + s$, and the saturation of the unitary bound implies the conservation of this operator $\partial \cdot J_s = \partial_{\lambda_1} J^{\lambda_1 \cdots \lambda_s} = 0$. For $s = 1$ and $s = 2$ the conserved currents generally exist, corresponding to the conservation of the global symmetry and stress tensor. In contrast, for $s > 2$ there are generically no conserved currents except for special cases, namely free theories and 2$d$ CFTs. The presence of conserved higher-spin currents in these special cases is a consequence of integrability [11–14]. Once the integrability is broken by an interaction, these higher-spin currents will acquire an anomalous dimension

$$\Delta_s = d - 2 + s + \gamma_s, \tag{1}$$

and subsequently a non-zero divergence,

$$\partial \cdot J_s = K_{s-1}. \tag{2}$$

These operators are called slightly broken higher-spin currents.

There are several motivations to study slightly broken higher-spin currents of interacting CFTs. Firstly, their anomalous dimensions $\gamma_s$ can serve as a measure of how interacting the theory is. Analysis based on the analytical bootstrap of four-point correlator predicts $\gamma_s$ to take a general form [15–21],

$$\gamma_s = c_1 \log s + c_2 + \mathcal{O}(1/s), \tag{3}$$

with $c_{1,2}$ being theory dependent numerical constants. Specifically, for theories with a perturbation parameter $\lambda$ (e.g. $\lambda \sim 1/N$ or $\lambda \sim \epsilon$ in the large-$N$ or $4 - \epsilon$ expansion) one has $c_{1,2} \sim \mathcal{O}(\lambda)$. Perturbatively, by evaluating the loop diagrams, $\gamma_s$ for O($N$) vector model [22] and Gross-Neveu-Yukawa theory [23] has been evaluated in the large-$N$ limit up to the order of $1/N^2$. Recently, a method utilising the classical equation of motion has been developed [24–29] and been applied to a variety of theories, such as Wilson-Fisher theory, cubic model, non-linear sigma models [24], Gross-Neveu-Yukawa model [26] and non-Abelian

Chern-Simons matter theories [25, 29]. It was found that $\gamma_s$ of the theories without gauge fields (i.e. Wilson-Fisher and Gross-Neveu-Yukawa) converges to a finite large-$s$ limit ($c_1 = 0$), while non-Abelian gauge theories have a logarithmically growing piece and was interpreted geometrically from the AdS/CFT correspondence [30].

Slightly broken higher-spin currents could also be useful for the conformal bootstrap using a recently proposed algorithm called hybrid bootstrap [31]. It has been observed that the performance of numerical bootstrap is largely improved if combined with the information of slightly broken higher-spin currents from analytical light-cone bootstrap. This also motivates us to study the slightly broken higher-spin currents of 3$d$ critical U(1) gauge theories, as it may help to bootstrap these CFTs.

These slightly broken higher-spin currents are also of theoretical importance as they impose strong constraints on interacting CFTs. For example, the divergence operator $K_{s-1}$ (2) has to be a spin $s-1$ primary operator with scaling dimension $\Delta = d - 1 + s$ in the free theory limit [32], and it becomes a descendent of $J_s$ in the interacting CFT. So the two different conformal multiplets of $J_s$ and $K_{s-1}$ in the free theories recombine into one when the interaction is turned on,

$$[J_s]_{\text{int.}} = [J_s]_{\text{free}} \cup [K_{s-1}]_{\text{free}}. \tag{4}$$

In other words, the shortening condition $\partial \cdot J_s = 0$ of the multiplet $[J_s]$ no longer hold in the interacting theory, and $[J_s]$ becomes a long multiplet by absorbing the multiplet $[K_{s-1}]$ in the free theory. This phenomenon is called conformal multiplet recombination [27, 33, 34], and it happens in all the interacting CFTs that can be accessed perturbatively via traditional renormalisation group. More interestingly, it was explicitly shown that one can define and calculate $d = 4 - \epsilon$ O($N$) Wilson-Fisher CFTs in a purely algebraic fashion using the conformal multiplet recombination [27].

An intriguing question one may raise is, is it possible to use the idea of conformal multiplet recombination to perform perturbative calculations that the renormalisation group is not applicable? One ideal target is 2$d$ CFTs,[1] which have conserved higher-spin currents due to the Virasoro symmetry. One would expect slightly broken higher-spin currents once 2$d$ CFTs are perturbed, for example, to $d = 2 + \epsilon$ dimensions. However, in 2$d$ CFTs' spectrum the divergence operators $K_{s-1}$ are absent,[2] making it elusive whether or not the idea of conformal multiplet recombination would work at all. In this paper, we resolve this mystery for 2$d$ Wess-Zumino-Witten (WZW) CFTs by reformulating them as gauge theories [36, 37]. The solution, as we will explain in the main text, is that $K_{s-1}$ is proportional to the gauge field strength, which happens to decouple from the IR spectrum in 2$d$. We then manage to calculate the leading order anomalous dimension of slightly broken higher-spin currents of SU($N$)$_1$ WZW CFTs at $2 + \epsilon$ dimensions.

In this paper, we apply the method based on the equation of motion to various U(1) gauge theories, including tricritical QED, scalar QED, fermionic QED, etc, and we find that the anomalous dimension $\gamma_s$ of the slightly broken higher-spin currents has similar logarithmic behaviour to the non-Abelian gauge theories studied before. This paper is organised as the following. In Section 2, we review the properties of the higher-spin currents in free theories, and the method to calculate their anomalous dimensions by making use of the equations of motion. The main result is also summarised in this section, and the details of the calculation are presented in the remaining sections. Specifically, in Section 3 we study bosonic QEDs in 3d, including the scalar QED and the tricritical QED, and in Section 4, we look at the fermionic QED and the QED-Gross-Neveu-Yukawa theory. In Section 5, we study the QED (i.e. SU($N$)$_1$ WZW) in $(2 + \epsilon)$-dimensions.

---

[1]See a recent attempt on the $2 + \epsilon$ Ising CFT [35].

[2]The statement is simply that in 2$d$ CFTs' spectrum, there is no global conformal primary (also called quasiprimary) with spin $s-1$ and scaling dimension $s + 1$.

## 2 Method and models

In this section we will introduce the slightly broken higher-spin currents in interacting theories and discuss the method to calculate their anomalous dimensions by making use of the classical equation of motion without calculating loop diagrams. This method was introduced in Ref. [24–26], and to be self-contained we also present its technical details. We will also review the models we study and present our main results.

### 2.1 The higher-spin currents in free field theory

To facilitate the description of higher-spin currents, we first introduce the index-free treatment of symmetric tensors [24,38]. For a rank-$l$ symmetric traceless tensor $T^{\lambda_1\cdots\lambda_l}$, we can contract it with an auxiliary polarisation vector $z_\lambda$,

$$\hat{T}_l \equiv T_{\lambda_1\cdots\lambda_s}z^{\lambda_1}\cdots z^{\lambda_s}\,. \tag{5}$$

By setting $z^\lambda$ to be null $z^2 = 0$, $\hat{T}_l$ only selects out the symmetric traceless part of the tensor. One can also go back to the full tensor by acting stripping operator on $\hat{T}_l$, which is a differential operator in $z$-space

$$D_z^\lambda \equiv \left(\frac{d}{2}-1\right)\partial_{z_\lambda} + z^\mu \partial_{z_\mu}\partial_{z_\lambda} - \frac{1}{2}z^\lambda \partial_{z_\mu}\partial_{z_\mu}\,. \tag{6}$$

Acting this operator once restores an index. By acting $D_z^\lambda$ repeatedly one can restore the uncontracted tensor

$$T_{\lambda_1\cdots\lambda_l} \propto D_{\lambda_1}^z \cdots D_{\lambda_l}^z \hat{T}_l\,. \tag{7}$$

In this description, the conservation of a spin-$s$ current $J_s^{\mu_1\cdots\mu_s}$ can be expressed concisely as

$$\partial \cdot \hat{J}_s = \partial_\mu D_z^\mu \hat{J}_s = 0\,. \tag{8}$$

By solving this equation, one can construct explicitly the conserved currents in free theories.

For the free theory of $N$-flavour complex scalar field $\phi^i$, $\phi^i$ transforms in the fundamental representation of the SU($N$) global symmetry. This theory is described by the Klein-Gordon Lagrangian

$$\mathscr{L}_0 = \partial_\mu \bar{\phi}_i \partial^\mu \phi^i\,, \tag{9}$$

and the scalar field satisfies the classical equation of motion

$$\partial^2 \phi^i = 0\,, \qquad \partial^2 \bar{\phi}_i = 0\,, \tag{10}$$

where $i = 1,\ldots,N$ and summation over repeated indices is implied. This free theory admits an infinite series of conserved higher-spin currents in the form of $s$ derivatives acting on the scalar bilinear $\bar{\phi}_j \phi^i -$ (trace) in the adjoint sector or $\bar{\phi}_k \phi^k$ in the singlet sector [24], i.e.

$$
\begin{aligned}
(\hat{J}_s^{(\mathrm{B})})^i{}_j(x) &= P_s^{(\mathrm{B})}(\hat{\partial}_1,\hat{\partial}_2)\bar{\phi}_j(x_1)\phi^i(x_2)\big|_{x_{1,2}\to x} - (\text{trace})\,, \\
\hat{J}_s^{(\mathrm{B})}(x) &= P_s^{(\mathrm{B})}(\hat{\partial}_1,\hat{\partial}_2)\bar{\phi}_k(x_1)\phi^k(x_2)\big|_{x_{1,2}\to x}\,,
\end{aligned}
\tag{11}
$$

where $\hat{\partial} = z^\lambda \partial_\lambda$. The trace substracted is $\frac{1}{d}\delta_j^i \hat{J}_s^{(\mathrm{B})}$, and

$$P^{(\mathrm{B})}(\xi,\eta) = \sum_{m=0}^{s} C_{sm}\xi^m \eta^{s-m} \tag{12}$$

is a homogeneous polynomial of degree $s$. By making use of the equations of motion Eq. (10), the conservation equation Eq. (8) reduces to a differential equation of the polynomial $P_s^{(B)}(\xi, \eta)$

$$\left(\frac{d}{2} - 1\right)\left[(P_s^{(B)})'_\xi(\xi, \eta) + (P_s^{(B)})'_\eta(\xi, \eta)\right] + \xi(P_s^{(B)})''_{\xi\xi}(\xi, \eta) + \eta(P_s^{(B)})''_{\eta\eta}(\xi, \eta) = 0. \tag{13}$$

Its solution can be expressed in terms of the order-$s$ Gegenbauer polynomial $C_s^\alpha$,

$$\begin{aligned}
P_s^{(B)}(\xi, \eta) &= (\xi + \eta)^s C_s^{(d-3)/2}\left(\frac{\xi - \eta}{\xi + \eta}\right) \\
&= \frac{\sqrt{\pi}\Gamma(\frac{d}{2} + s - 1)\Gamma(d + s - 3)}{2^{d-4}\Gamma\left(\frac{d-3}{2}\right)} \sum_{m=0}^{s} \frac{(-1)^m \xi^m \eta^{s-m}}{m!(s-m)!\Gamma(m + \frac{d}{2} - 1)\Gamma(s - m + \frac{d}{2} - 1)}.
\end{aligned} \tag{14}$$

Note that this expression only holds for $d \neq 3$. For $d = 3$, it vanishes due to the improperly chosen renormalisation factor, and one can use instead (it only differs from Eq. (14) by a factor)

$$P_s^{(B)}(\xi, \eta) = \frac{(\sqrt{\xi} + i\sqrt{\eta})^{2s} + (\sqrt{\xi} - i\sqrt{\eta})^{2s}}{2 \cdot s!}. \tag{15}$$

For the free theory of $N$-flavour fermionic field $\psi^i$, $\psi^i$ transforms in the fundamental representation of a SU($N$) global symmetry. This theory is described by the Dirac Lagrangian

$$\mathcal{L}_0 = -\bar{\psi}_i \slashed{\partial} \psi^i, \tag{16}$$

and the scalar field satisfies the classical equation of motion

$$\slashed{\partial}\psi^i = 0, \qquad \partial_\mu \bar{\psi}_i \gamma^\mu = 0. \tag{17}$$

This free theory admits an infinite series of conserved higher-spin currents in the form of $(s-1)$ derivatives acting on the fermion bilinear $\bar{\psi}_j \hat{\gamma} \psi^i - $ (trace) in the adjoint sector or $\bar{\psi}_k \hat{\gamma} \psi^k$ in the singlet sector [25], i.e.

$$\begin{aligned}
(\hat{J}_s^{(F)}(x))^i{}_j &= P_s^{(F)}(\hat{\partial}_1, \hat{\partial}_2)\bar{\psi}_j(x_1)\hat{\gamma}\psi^i(x_2)\big|_{x_{1,2} \to x} - \text{(trace)}, \\
\hat{J}_s^{(F)}(x) &= P_s^{(F)}(\hat{\partial}_1, \hat{\partial}_2)\bar{\psi}_j(x_1)\hat{\gamma}\psi^i(x_2)\big|_{x_{1,2} \to x},
\end{aligned} \tag{18}$$

where

$$P^{(F)}(\xi, \eta) = \sum_{k=0}^{s-1} C_{sk}\xi^k \eta^{s-k}. \tag{19}$$

The conservation asserts that $P_s^{(F)}$ should satisfy

$$\frac{d}{2}\left[(P_s^{(F)})'_\xi(\xi, \eta) + (P_s^{(F)})'_\eta(\xi, \eta)\right] + \xi(P_s^{(F)})''_{\xi\xi}(\xi, \eta) + \eta(P_s^{(F)})''_{\eta\eta}(\xi, \eta) = 0. \tag{20}$$

Its solution can be expressed in terms of the order-$(s-1)$ Gegenbauer polynomial,

$$\begin{aligned}
P_s^{(F)}(\xi, \eta) &= (\xi + \eta)^{s-1} C_{s-1}^{(d-1)/2}\left(\frac{\xi - \eta}{\xi + \eta}\right) \\
&= \frac{\sqrt{\pi}\Gamma(\frac{d}{2} + s - 1)\Gamma(d + s - 2)}{2^{d-1}\Gamma(\frac{d-1}{2})} \sum_{m=0}^{s-1} \frac{(-1)^m \xi^m \eta^{s-1-m}}{m!(s-1-m)!\Gamma(m + \frac{d}{2})\Gamma(s - m + \frac{d}{2})}.
\end{aligned} \tag{21}$$

## 2.2 Anomalous dimensions of the slightly broken higher-spin currents

Now let us turn on adiabatically an interaction characterised by some small expansion parameter $\lambda$. For example, in the large-$N$ limit, one can take $\lambda = 1/N$. The previously conserved currents will evolve into some higher-spin operators which are no longer conserved for general $\lambda$ which we call 'slightly broken higher-spin currents' with scaling dimension

$$\Delta_s = d - 2 + s + \gamma_s, \tag{22}$$

where the anomalous dimension $\gamma_s$ is at least of order $\mathcal{O}(\lambda)$. Correspondingly, the divergence of the operators

$$\partial \cdot \hat{J}_s = \hat{K}_{s-1} \tag{23}$$

are generally non-zero and of order $\mathcal{O}(\lambda)$. We reiterate that this equation describes the physics of conformal multiplet recombination, namely the short multiplet of $J_s$ and the long multiplet of $K_{s-1}$ in the free limit recombines into a long multiplet of $J_s$ in the interacting theory. For this equation to hold, a necessary condition is that $K_{s-1}$ is a primary operator with spin $s-1$ and scaling dimension $\Delta = d - 1 + s$.

The divergence operator $K_{s-1}$ can generally be fixed by making use of the equation of motion. For example, for a scalar field, the equation of motion usually takes the form $\partial^2 \phi = \lambda V$, where $V$ is some spin-0 and engineering dimension-$(\frac{d}{2} + 1)$ operator. The expression for $\hat{K}_{s-1}$ can be obtained by substituting $\partial^2 \phi$ by $\lambda V$ repeatedly in $\partial \cdot \hat{J}_s$. As we will see later explicitly, for $s = 1$ adjoint and $s = 2$ singlet sector, we have $\partial \cdot (J_1)^i{}_j = 0$ and $\partial \cdot J_2 = 0$, corresponding to the conservation of the symmetry current and stress tensor.

Eq. (23) can be used to calculate the anomalous dimension. We write down the generic form of the two-point correlator of the spin-$s$ current

$$\left\langle \hat{J}_s(x) \hat{J}_s(x') \right\rangle = C^{(\text{sg})} \frac{\left( z \cdot z' - 2 \frac{(z \cdot X)(z' \cdot X)}{X^2} \right)^s}{X^{2\Delta_s^{(\text{sg})}}},$$

$$\left\langle (\hat{J}_s)^i{}_j(x)(\hat{J}_s)^k{}_l(x') \right\rangle = C^{(\text{ad})} \frac{\left( z \cdot z' - 2 \frac{(z \cdot X)(z' \cdot X)}{X^2} \right)^s}{X^{2\Delta_s^{(\text{ad})}}} \left( \delta^i_l \delta^k_j - \frac{1}{N} \delta^i_j \delta^k_l \right), \tag{24}$$

where $X = x - x'$. For simplicity, we will discuss only the singlet sector in the following, and the adjoint sector follows similarly. We take the divergence with respect to both $x$ and $x'$ and then set $z = z'$ [24]

$$\left\langle \hat{K}_{s-1}(x) \hat{K}_{s-1}(x') \right\rangle \big|_{z'=z} = \partial_\mu D_z^\mu \partial'_\nu D_{z'}^\nu \left\langle \hat{J}_s(x) \hat{J}_s(x') \right\rangle \big|_{z'=z}$$

$$= -\frac{1}{\hat{X}^2} s \left( s + \frac{d}{2} - 2 \right) \left[ \gamma_s \left( s + \frac{d}{2} - 1 \right) (s + d - 3) \right.$$

$$\left. + \gamma_s^2 \left( s^2 + \left( \frac{d}{2} - 2 \right) s + \frac{d}{2} - 1 \right) \right] \left\langle \hat{J}_s(x) \hat{J}_s(x') \right\rangle. \tag{25}$$

Note that the second line is proportional to $\gamma_s$. This results from the higher-spin current conservation at zero coupling. By evaluating both sides to the leading order, we can obtain the expression for the anomalous dimension

$$\gamma_s = -\frac{1}{s \left( s + \frac{d}{2} - 2 \right) \left( s + \frac{d}{2} - 1 \right) (s + d - 3)} \frac{\hat{X}^2 \left\langle \hat{K}_{s-1}(x) \hat{K}_{s-1}(x') \right\rangle_{(\text{leading order})}}{\left\langle \hat{J}_s(x) \hat{J}_s(x') \right\rangle_{(\text{leading order})}} + \mathcal{O}(\lambda^2). \tag{26}$$

## 2.3 Models and summary of results

In this paper, we apply Eq. (26) to various gauge theories to calculate the anomalous dimension of the slightly broken higher-spin currents.

We first discuss the tricritical QED and scalar QED in the large-$N$ limit [39,40]. For scalar fields, one can couple the free theory to a gauge field by replacing the derivatives in the free Lagrangian (9) with covariant derivatives

$$\mathscr{L}_e = D_\mu \bar{\phi}_i D^\mu \phi^i + \frac{1}{4e^2} F_{\mu\nu} F^{\mu\nu}, \tag{27}$$

where $D_\mu \phi^i = (\partial_\mu - iA_\mu)\phi^i$ and $D_\mu \bar{\phi}_i = (\partial_\mu + iA_\mu)\bar{\phi}_i$. This theory admits another relevant operator, namely the quartic coupling $\frac{\lambda}{4N}(\bar{\phi}_i \phi^i)^2$. This term can be written equivalently in terms of a Hubbard-Stratonovich field $\sigma$,

$$\mathscr{L}_\sigma = \sigma \bar{\phi}_i \phi^i - \frac{N}{4\lambda}\sigma^2. \tag{28}$$

This theory flows a fixed point in the infrared corresponding to an interacting CFT called the scalar QED. In the large-$N$ limit, one can also tune $\lambda = 0$ to get a different CFT called the tricritical QED in the infrared.

As will be discussed in Section 3, the anomalous dimensions of slightly broken higher-spin currents of these two theories in $3d$ are,

$$\gamma_s^{\text{tricr.QED,ad}} = \frac{16}{N\pi^2}\left[\sum_{i=1}^s \frac{1}{i-1/2} - \frac{2(11s^2-2)}{3(4s^2-1)}\right],$$

$$\gamma_s^{\text{tricr.QED,sg}} = \frac{16}{N\pi^2}\left[\sum_{i=1}^s \frac{1}{i-1/2} - \begin{cases} \dfrac{2(11s^4+3s^3-13s^2+15s+2)}{3(s^2-1)(4s^2-1)}, & s \text{ even} \\ \dfrac{2(11s^2-2)}{3(4s^2-1)}, & s \text{ odd} \end{cases}\right],$$

$$\gamma_s^{\text{scal.QED,ad}} = \frac{16}{N\pi^2}\left[\sum_{i=1}^s \frac{1}{i-1/2} - \frac{7s^2-1}{4s^2-1}\right],$$

$$\gamma_s^{\text{scal.QED,sg}} = \frac{16}{N\pi^2}\left[\sum_{i=1}^s \frac{1}{i-1/2} - \begin{cases} \dfrac{14s^4+5s^3-16s^2+19s+2}{2(s^2-1)(4s^2-1)}, & s \text{ even} \\ \dfrac{7s^3+2s^2-s-2}{s(4s^2-1)}, & s \text{ odd} \end{cases}\right], \tag{29}$$

where 'ad' and 'sg' denote the currents in the adjoint and singlet sector. To the first order of large-$N$ expansion, in the limit $s \to \infty$, the asymptotic behaviour is

$$\gamma_s \sim \frac{16}{N\pi^2}\left[\log s - \gamma - 2\log 2 - \begin{cases} \frac{11}{6} & \text{tricr.QED} \\ \frac{7}{4} & \text{scal.QED} \end{cases} + \mathcal{O}\left(\frac{1}{s}\right)\right], \tag{30}$$

where $\gamma$ is the Euler Gamma constant. Note that the asymptotic behaviours are the same for singlet and adjoint sector. We have also discussed the bosonic QEDs with Chern-Simons terms. The results are included in Appendix B. Note $\gamma_{s=1}^{\text{ad}} = 0$ and $\gamma_{s=2}^{\text{sg}} = 0$, corresponding to the conservation of $SU(N)$ symmetry current and energy momentum tensor.

We can similarly define the $\text{QED}_3$ and $\text{QED}_3$-Gross-Neveu-Yukawa theories for fermion fields[3] in the large-$N$ limit [39–43]. One can couple the free fermionic theory to a gauge field by replacing the derivatives in the free Lagrangian (16) with covariant derivatives

$$\mathscr{L}_e = -\bar{\psi}_i \slashed{D} \psi^i, \tag{31}$$

---

[3]The fermion flavor number is counted in the unit of 2-component fermions.

where $D_\mu \psi = (\partial_\mu - iA_\mu)\psi$, and the irrelevant Maxwell term is omitted. We find the anomalous dimensions of the slightly broken higher-spin currents of fermionic QED are exactly the same as in tricritical QED, its bosonic counterpart.[4] The detailed calculation is enclosed in Section 4.

One can also couple the fermions mass to a critical bosonic field through a Gross-Neveu-Yukawa interaction

$$\mathscr{L}_\sigma = \sigma \bar\psi_i \psi^i. \tag{32}$$

Again, we find the anomalous dimensions of the slightly broken higher-spin currents of fermionic QED with GNY interaction are exactly the same as in scalar QED. The detailed calculation is enclosed in Section 4.

Another particularly interesting limit is the QED (i.e. $SU(N)_1$ WZW CFT) in $(2+\epsilon)$-dimension, which is discussed in Section 5, and the anomalous dimensions in the adjoint sector are calculated to be

$$\gamma_s^{\text{QED,ad}} = \frac{\epsilon}{N}\sum_{i=1}^{s-1}\frac{1}{i} + \mathscr{O}\left(\frac{\epsilon^2}{N}\right) = \frac{\epsilon}{N}H_{s-1} + \mathscr{O}\left(\frac{\epsilon^2}{N}\right), \tag{33}$$

where $H_{s-1}$ is the Harmonic number.

## 3 Bosonic QEDs in 3$d$ large-$N$ limit

In this section, we calculate the anomalous dimension of the tricritical QED and scalar QED in the large-$N$ limit, with Lagrangian defined in Eqs. (27) and (28). In the large-$N$ limit, the scalar field bubble diagrams can be resummed to be an effective photon propagator. In $d < 4$, the Maxwell term is irrelevant and can be omitted. In other words, we take the $e^2 \to \infty$ limit to do the calculation from the beginning. A non-local gauge fixing described in Ref. [40,41] is adopted. Similarly, in the scalar QED, the effective propagator of $\sigma$ is obtained by resumming the bubble diagrams [44], and its mass term can be omitted. The scalar field propagator in the large-$N$ limit is the same as in free field theory. The Feynman rules altogether are listed below:

$$G^i{}_j(x) = \langle \phi^i(x)\phi_j(0)\rangle_\infty = \delta^i_j \frac{\Gamma\left(\frac{d}{2}-1\right)}{4\pi^{d/2}}\frac{1}{x^{d-2}},$$

$$D^{\mu\nu}(x) = \langle A^\mu(x)A^\nu(0)\rangle_\infty = \frac{1}{N}\frac{-\Gamma(d)\sin\frac{\pi d}{2}}{\pi(d-2)\Gamma\left(\frac{d}{2}\right)^2}\frac{(d-2-\zeta)\delta^{\mu\nu}+2\zeta\frac{x^\mu x^\nu}{x^2}}{x^2}, \tag{34}$$

$$D_\sigma(x) = \langle \sigma(x)\sigma(0)\rangle_\infty = \frac{1}{N}\frac{8(d-4)\Gamma(d-2)\sin\frac{\pi d}{2}}{\Gamma(\frac{d}{2}-1)^2}\frac{1}{x^4}.$$

### 3.1 Tricritical QED currents

The equations of motion for $\phi$, $\bar\phi$ and $A$ that we will make use of are

$$\partial^2\phi^i = i(\partial_\mu A^\mu)\phi^i + 2iA^\mu(\partial_\mu\phi^i) + \mathscr{O}(A^2), \tag{35a}$$

$$\partial^2\bar\phi_j = -i\bar\phi_j(\partial_\mu A^\mu) - 2i(\partial_\mu\bar\phi_j)A^\mu + \mathscr{O}(A^2), \tag{35b}$$

$$0 = i(\partial_{1\mu} - \partial_{2\mu})\bar\phi_k(x_1)\phi^k(x_2)\big|_{x_{1,2}\to x} + \mathscr{O}(A). \tag{35c}$$

---

[4]Similar coincidence has also been found between 3$d$ $O(N)$ Wilson-Fisher and Gross-Neveu-Yukawa theory [22, 23]. This coincidence is not an indication of any type of duality, and will no longer hold at the $1/N^2$ order.

We keep these equations to the leading order in $A$, as the correlation function of $A$ is a small quantity of order $1/N$.

When the coupling to the gauge field is turned on, the slightly broken higher-spin currents should be modified to be gauge invariant by replacing the partial derivative in Eq. (11) with covariant derivative [25].

$$\hat{J}_s^{(\mathrm{B})}(x) = P_s^{(\mathrm{B})}(\hat{D}_1, \hat{D}_2)\bar{\phi}(x_1)\phi(x_2)\big|_{x_{1,2}\to x}, \tag{36}$$

where the polynomial $P_s^{(\mathrm{B})}(\xi, \eta)$ is given in Eq. (14) and (15). Here we omit the flavour index. This expression applies both to singlet and adjoint sector. To the leading order in $1/N$, we need only to expand the expression to the linear order in the gauge field. When acting a power of $\hat{D}$ on $\phi$, one gets

$$\hat{D}^n\phi(x) = (\hat{\partial} - i\hat{A})^n\phi(x) = \hat{\partial}^n\phi(x) - i\sum_{m=0}^{n-1}\hat{\partial}^m\hat{A}\hat{\partial}^{n-1-m}\phi + \mathcal{O}(A^2)$$

$$= \hat{\partial}^n\phi(x) - i\frac{(\hat{\partial} + \hat{\partial}')^n - \hat{\partial}^n}{\hat{\partial}'}\hat{A}(x')\phi(x)\bigg|_{x'\to x} + \mathcal{O}(A^2). \tag{37}$$

Generalise this expression to a polynomial of $\hat{D}$, one obtains $\hat{J}_s(x)$ to the linear order of $\hat{A}$

$$\hat{J}_s(x) = P(\hat{\partial}_1, \hat{\partial}_2)\bar{\phi}(x_1)\phi(x_2) + iQ(\hat{\partial}_1, \hat{\partial}_2, \hat{\partial}_3)\bar{\phi}(x_1)\hat{A}(x_3)\phi(x_2)\big|_{x_{1,2,3}\to x} = \hat{J}_s^{(P)}(x) + \hat{J}_s^{(Q)}(x), \tag{38}$$

where

$$Q(\xi, \eta, \chi) = \frac{P(\xi + \chi, \eta) - P(\xi, \eta + \chi)}{\chi}. \tag{39}$$

We then calculate its divergence

$$\hat{K}_{s-1} = \partial_\mu D_z^\mu \hat{J}_s. \tag{40}$$

The divergence of the $A$-independent part $\hat{J}_s^{(P)}$ is

$$\hat{K}_{s-1}^{(P)} = \left[M_1(\hat{\partial}_1, \hat{\partial}_2)\partial_1^2 + M_2(\hat{\partial}_1, \hat{\partial}_2)\partial_2^2\right]\bar{\phi}(x_1)\phi(x_2), \tag{41}$$

where

$$M_1(\xi, \eta) = \frac{d-2}{2}P_\xi'(\xi, \eta) + \frac{\xi - \eta}{2}P_{\xi\xi}''(\xi, \eta) + \eta P_{\xi\eta}''(\xi, \eta),$$

$$M_2(\xi, \eta) = \frac{d-2}{2}P_\eta'(\xi, \eta) - \frac{\xi - \eta}{2}P_{\eta\eta}''(\xi, \eta) + \xi P_{\xi\eta}''(\xi, \eta). \tag{42}$$

We then substitute in the equations of motion Eq. (35a) and (35b). The divergence of the $A$-dependent part $\hat{J}_s^{(Q)}$ can be calculated similarly. For this part, one can simply use the free equation of motion $\partial^2\phi = 0$. Combined together, the final result is gauge invariant, dependent only on the field strength $F_{\mu\nu} = \partial_\mu A_\nu - \partial_\nu A_\mu$.

$$\hat{K}_{s-1} = \mathscr{K}_{(A)}^{\mu\nu}(\partial_1, \partial_2, \partial_3)i\bar{\phi}(x_1)\phi(x_2)F_{\mu\nu}(x_3)\big|_{x_{1,2,3}\to x}, \tag{43}$$

where the differential operator

$$\mathscr{K}_{(A)}^{\mu\nu}(\partial_1, \partial_2, \partial_3) = R_1(\hat{\partial}_1, \hat{\partial}_2, \hat{\partial}_3)\partial_1^\mu z^\nu + R_2(\hat{\partial}_1, \hat{\partial}_2, \hat{\partial}_3)\partial_2^\mu z^\nu + R_3(\hat{\partial}_1, \hat{\partial}_2, \hat{\partial}_3)\partial_3^\mu z^\nu, \tag{44}$$

and the polynomials

$$R_1(\xi, \eta, \chi) = \frac{2}{\chi} M_1(\xi + \chi, \eta) - \frac{1}{\chi} \left[ \frac{d-2}{2} Q - (\eta + \chi) Q'_\xi + \eta Q'_\eta + \chi Q'_\chi \right],$$

$$R_2(\xi, \eta, \chi) = -\frac{2}{\chi} M_2(\xi, \eta + \chi) - \frac{1}{\chi} \left[ \frac{d-2}{2} Q + \xi Q'_\xi - (\eta + \chi) Q'_\eta + \chi Q'_\chi \right], \quad (45)$$

$$R_3(\xi, \eta, \chi) = \frac{1}{\chi} [M_1(\xi + \chi, \eta) - M_2(\xi, \eta + \chi)] - \frac{1}{\chi} \left[ \frac{d-2}{2} Q + \xi Q'_\xi + \eta Q'_\eta - (\xi + \eta) Q'_\chi \right].$$

Especially, in $3d$, for $s = 1$ adjoint, $(\hat{K}_0)^i{}_j = 0$, and for $s = 2$ singlet, due to the equation of motion (35c) for $A$, $\hat{K}_1 = 6i(\partial_1^\mu - \partial_2^\mu)\bar{\phi}_{k,1}\phi_2^k F_{\mu\nu,3} z^\nu = 0$, which corresponds to the conservation of symmetry current and stress tensor.

### 3.1.1  Adjoint sector

We first calculate the anomalous dimension in the adjoint sector. To do this, we restore the flavour index for the slightly broken higher-spin currents

$$(\hat{J}_s)^i{}_j = P(\hat{\partial}_1, \hat{\partial}_2)\bar{\phi}_{j,1}\phi_2^i + iQ(\hat{\partial}_1, \hat{\partial}_2, \hat{\partial}_3)i\bar{\phi}_{j,1}\phi_2^i \hat{A}_3 - (\text{trace}), \quad (46)$$

and its divergence

$$(\hat{K}_{s-1})^i{}_j = \mathscr{K}^{\mu\nu}_{(A)}(\partial_1, \partial_2, \partial_3) i \bar{\phi}_{j,1} \phi_2^i F_{\mu\nu,3} - (\text{trace}), \quad (47)$$

and then calculate their correlation function in the large-$N$ limit and plug it into Eq. (26).

For the correlation of $\hat{J}_s$, the contribution of $\hat{A}$-dependent piece is of higher order and thus can be omitted, leaving only the $\hat{A}$-independent part. In the adjoint sector, the only contribution to the bilinear correlator is the direct contraction, thus

$$\langle (\hat{J}_s)^i{}_j(x)(\hat{J}_s)^k{}_l(0) \rangle = P(\hat{\partial}_1, \hat{\partial}_2)P(\hat{\partial}_{1'}, \hat{\partial}_{2'}) \langle (\overline{\bar{\phi}_{1,j}\phi_2^i})(\bar{\phi}'_{1,k}\phi_2'^l) \rangle \Big|_{x_{1,2} \to x, x'_{1,2} \to 0} - (\text{trace})$$

$$= P(\hat{\partial}_1, \hat{\partial}_2)P(-\hat{\partial}_2, -\hat{\partial}_1)G^k{}_j(x_1)G^i{}_l(x_2)\Big|_{x_{1,2} \to x} - (\text{trace}). \quad (48)$$

This expression can be evaluated by using the Schwinger parametrisation of the propagator [24]

$$G^i{}_j(x) = \delta^i_j \int_0^\infty \frac{d\alpha}{4\pi^{d/2}} \alpha^{d/2-2} e^{-\alpha x^2}. \quad (49)$$

When acting on the integrand, the hatted differential operators $\hat{\partial}$ can be replaced by $-2\alpha\hat{x}$, due to the null condition $z^2 = 0$ and subsequently $\hat{\partial}\hat{x} = 0$. Hence,

$$\langle (\hat{J}_s)^i{}_j(x)(\hat{J}_s)^k{}_l(0) \rangle = \left( \delta^i_l \delta^k_j - \frac{1}{N} \delta^i_j \delta^k_l \right)$$

$$\times \int_0^\infty \int_0^\infty \frac{d\alpha_1}{4\pi^{d/2}} \frac{d\alpha_2}{4\pi^{d/2}} P(-2\alpha_1\hat{x}, -2\alpha_2\hat{x})P(2\alpha_2\hat{x}, 2\alpha_1\hat{x})\alpha_1^{d/2-2}\alpha_2^{d/2-2} e^{-(\alpha_1+\alpha_2)x^2}. \quad (50)$$

Similar techniques can be used to evaluate the correlation function of $\hat{K}_{s-1}$

$$\langle (\hat{K}_{s-1})^i{}_j(x)(\hat{K}_{s-1})^k{}_l(0) \rangle = -\mathscr{K}^{\mu\nu}_{(A)}(\partial_1, \partial_2, \partial_3)\mathscr{K}^{\mu\nu}_{(A)}(-\partial_2, -\partial_1, \partial_3)$$

$$\times G^k{}_j(x_1)G^i{}_l(x_2)D^{\mu\nu,\rho\sigma}_P(x_3)\Big|_{x_{1,2,3} \to x} - (\text{trace}), \quad (51)$$

where $D_P^{\mu\nu,\rho\sigma}(x) = \langle F^{\mu\nu}(x)F^{\rho\sigma}(0)\rangle_\infty$.

It is difficult to evaluate the Schwinger integral for Eqs. (48) and (51) for general $s$, so instead, we evaluate the expression for finite $s$ in $3d$ up to $s = 50$ and then match it with an analytic expression. Our final result is

$$\gamma_s^{\text{tricr.QED,ad}} = \frac{16}{N\pi^2}\left[\sum_{i=1}^{s}\frac{1}{i-1/2} - \frac{2(11s^2-2)}{3(4s^2-1)}\right]. \tag{52}$$

### 3.1.2 Singlet sector

In the singlet sector, the higher-spin operator $\hat{J}_s$ and its divergence $\hat{K}_{s-1}$ are

$$\begin{aligned}\hat{J}_s &= P(\hat{\partial}_1, \hat{\partial}_2)\bar{\phi}_{j,1}\phi_2^i + iQ(\hat{\partial}_1, \hat{\partial}_2, \hat{\partial}_3)i\bar{\phi}_{k,1}\phi_2^k\hat{A}_3,\\\hat{K}_{s-1} &= \mathcal{K}_{(A)}^{\mu\nu}(\partial_1, \partial_2, \partial_3)i\bar{\phi}_{k,1}\phi_2^kF_{\mu\nu,3}.\end{aligned} \tag{53}$$

Note that the correlators of $\hat{J}_s$ and $\hat{K}_{s-1}$ are no longer only contractions. To the leading order of $1/N$, we have to consider other possible contributions. Here we outline the process of calculation and enclose the details in Appendix A.

For the correlation of $\hat{J}_s$, It can be shown that the $s = 1$ current drops out from the spectrum, and for $s \geq 2$, the only contribution to its correlation function is still direct contraction. For the correlation of $\hat{K}_s$, adding extra contribution is equivalent to making use of the equation of motion (35c) for $A_\mu$ which effectively remove the pieces proportional to $\hat{J}_1$. More specifically, we rewrite the divergence in terms of the slightly broken higher-spin currents, the field strength and their descendants.

$$\hat{K}_{s-1} = \sum_{l=0}^{s-2}[J_l][F], \tag{54}$$

where $[\dots]$ denotes the conformal family of the operator, and in $J_l$ we need only to keep the $\hat{A}$-independent piece. The correlation of $\hat{K}_{s-1}$ in large-$N$ limit can be written effectively as the contraction of $\hat{\tilde{K}}_{s-1}$ in which the terms proportional to $J_1$ are removed from $\hat{K}_{s-1}$.

$$\left\langle\hat{K}_{s-1}(x)\hat{K}_{s-1}(0)\right\rangle_\infty = \left\langle\hat{\tilde{K}}_{s-1}(x)\hat{\tilde{K}}_{s-1}(0)\right\rangle_{\text{ct.}}, \qquad \hat{\tilde{K}}_{s-1} = \hat{K}_{s-1} - [J_1][F]. \tag{55}$$

Taking this extra contribution into account, we evaluate the anomalous dimension for finite $s$ and extrapolate an analytic expression in $3d$. Our final result is

$$\gamma_s^{\text{tricr.QED,sg}} = \frac{16}{N\pi^2}\left[\sum_{i=1}^{s}\frac{1}{i-1/2} - \begin{cases}\dfrac{2(11s^4+3s^3-13s^2+15s+2)}{3(s^2-1)(4s^2-1)}, & s\text{ even}\\[2mm]\dfrac{2(11s^2-2)}{3(4s^2-1)}, & s\text{ odd}\end{cases}\right]. \tag{56}$$

## 3.2 Scalar QED

In this section we consider the scalar QED, with Lagrangian defined in Eqs. (27) and (28). The modified equation of motion for $\phi$ are, to linear order of $A$ and $\sigma$

$$\begin{aligned}\partial^2\phi^i &= i(\partial_\mu A^\mu)\phi^i + 2iA^\mu(\partial_\mu\phi^i) + \phi^i\sigma,\\\partial^2\bar{\phi}_j &= -i\bar{\phi}_j(\partial_\mu A^\mu) - 2i(\partial_\mu\bar{\phi}_j)A^\mu + \bar{\phi}_j\sigma.\end{aligned} \tag{57}$$

and there is an additional equation of motion for $\sigma$

$$0 = \bar{\phi}_k\phi^k. \tag{58}$$

Substitute the equation of motion (57) into the divergence Eq. (42), we get an extra piece in the divergence $\hat{K}_{s-1}$.

$$\hat{K}_{s-1} = \mathcal{K}_{(A)}^{\mu\nu}(\partial_1, \partial_2, \partial_3) i\bar{\phi}(x_1)\phi(x_2)F_{\mu\nu}(x_3) + \mathcal{K}_{(\sigma)}(\partial_1, \partial_2, \partial_3) i\bar{\phi}(x_1)\phi(x_2)\sigma(x_3)\Big|_{x_{1,2,3}\to x}, \tag{59}$$

where

$$\mathcal{K}_{(\sigma)} = M_1(\hat{\partial}_1 + \hat{\partial}_3, \hat{\partial}_2) + M_2(\hat{\partial}_1, \hat{\partial}_2 + \hat{\partial}_3). \tag{60}$$

In the adjoint sector, the correlation of the higher-spin operator $\hat{J}_s$ remains the same as the tricritical QED, and the correlation of the divergence $\hat{K}_{s-1}$ can be evaluated by direct contraction,

$$\langle (\hat{K}_{s-1})^i_{\ j}(x)(\hat{K}_{s-1})^k_{\ l}(0)\rangle = -\mathcal{K}_{(A)}^{\mu\nu}(\partial_1, \partial_2, \partial_3)\mathcal{K}_{(A)}^{\mu\nu}(-\partial_2, -\partial_1, \partial_3)G^k_{\ j}(x_1)G^i_{\ l}(x_2)D_P^{\mu\nu,\rho\sigma}(x_3)$$
$$+\mathcal{K}_{(\sigma)}(\partial_1, \partial_2, \partial_3)\mathcal{K}_{(\sigma)}(-\partial_2, -\partial_1, \partial_3)G^k_{\ j}(x_1)G^i_{\ l}(x_2)D_\sigma(x_3)\Big|_{x_{1,2,3}\to x} - (\text{trace}). \tag{61}$$

Substitute this into Eq. (26), we get the result for the anomalous dimension in $3d$

$$\gamma_s^{\text{scal.QED,ad}} = \frac{16}{N\pi^2}\left(\sum_{i=1}^s \frac{1}{i-1/2} - \frac{7s^2-1}{4s^2-1}\right). \tag{62}$$

In the singlet sector, for the correlation of the divergence $\hat{K}_{s-1}$, we need to take into account the equation of motion (35c) for $A$ and (58) for $\sigma$. We write $\hat{K}_{s-1}$ in terms of $J_l$, $F$, $\sigma$ and their descendents

$$\hat{K}_{s-1} = \sum_{l=0}^{s-2}[J_l][F] + \sum_{l=0}^{s-1}[J_l][\sigma] \tag{63}$$

remove the pieces proportional to $J_1$ and $J_0$,

$$\hat{\tilde{K}}_{s-1} = \hat{K}_{s-1} - [J_1][F] - [J_0][F] - [J_1][\sigma] - [J_0][\sigma], \tag{64}$$

and calculate its correlator through direct contraction

$$\langle \hat{K}_{s-1}(x)\hat{K}_{s-1}(0)\rangle_\infty = \langle \hat{\tilde{K}}_{s-1}(x)\hat{\tilde{K}}_{s-1}(0)\rangle_{\text{ct.}}. \tag{65}$$

The result for the anomalous dimension in $3d$ is

$$\gamma_s^{\text{scal.QED,sg}} = \frac{16}{N\pi^2}\left[\sum_{i=1}^s \frac{1}{i-1/2} - \begin{cases} \dfrac{14s^4+5s^3-16s^2+19s+2}{2(s^2-1)(4s^2-1)}, & s \text{ even} \\[2ex] \dfrac{7s^3+2s^2-s-2}{s(4s^2-1)}, & s \text{ odd} \end{cases}\right]. \tag{66}$$

# 4 Fermionic QEDs in 3$d$ large-$N$ limit

In this section we calculate the anomalous dimension of the QED$_3$ and QED$_3$-Gross-Neveu-Yukawa theory in the large-$N$ limit, with Lagrangian defined in Eqs. (31) and (32). The effective photon propagator in the large-$N$ limit comes from the resummation of the fermion bubble diagrams. The large-$N$ fermion propagator is the same as in free field theory. The Feynman rules of QED$_3$ are listed below:

$$G^i_{\ j}(x) = \langle \psi^i(x)\bar{\psi}_j(0)\rangle_\infty = \delta^i_j \frac{\Gamma\left(\frac{d}{2}\right)}{2\pi^{d/2}}\frac{\slashed{x}}{x^d},$$
$$D^{\mu\nu}(x) = \langle A^\mu(x)A^\nu(0)\rangle_\infty = \frac{1}{N}\frac{-\Gamma(d)\sin\frac{\pi d}{2}}{\pi(d-2)\Gamma\left(\frac{d}{2}\right)^2}\frac{(d-2-\zeta)\delta^{\mu\nu}+2\zeta\frac{x^\mu x^\nu}{x^2}}{x^2}. \tag{67}$$

For QED$_3$-Gross-Neveu-Yukawa theory there is one extra Feynman rule for the effective propagator of $\sigma$ obtained by resumming the fermion bubble diagrams,

$$D_\sigma(x) = \langle \sigma(x)\sigma(0)\rangle_\infty = \frac{1}{N}\frac{2^{d-1}\Gamma(\frac{d-1}{2})\sin\frac{\pi d}{2}}{\pi^{3/2}\Gamma(\frac{d}{2}-1)^2}\frac{1}{x^2}\,. \tag{68}$$

## 4.1 QED$_3$

The equations of motion for $\psi$ and $\bar\psi$ to the linear order in $A$ are

$$\slashed{\partial}\psi = i\slashed{A}\psi\,,$$
$$\partial_\mu\bar\psi\gamma^\mu = -i\bar\psi\slashed{A}\,,$$
$$\partial^2\psi = \frac{i}{2}\gamma_\lambda\epsilon^{\mu\nu\lambda}F_{\mu\nu}\psi + i(\partial_\mu A^\mu)\psi + 2iA^\mu(\partial_\mu\psi)\,,$$
$$\partial^2\bar\psi = \frac{i}{2}\bar\psi F_{\mu\nu}\gamma_\lambda\epsilon^{\mu\nu\lambda} - i\bar\psi(\partial_\mu A^\mu) - 2i(\partial_\mu\bar\psi)A^\mu\,, \tag{69}$$

where we have made use of the $\gamma$-matrix identity

$$\gamma^\mu\gamma^\nu = \delta^{\mu\nu} + i\epsilon^{\mu\nu\lambda}\gamma_\lambda\,. \tag{70}$$

The gauge invariant slightly broken higher-spin currents are

$$\hat J_s^{(F)}(x) = P_s^{(F)}(\hat D_1, \hat D_2)\bar\psi(x_1)\hat\gamma\psi(x_2)\big|_{x_{1,2}\to x}\,, \tag{71}$$

where the polynomial $P_s^{(F)}(\xi, \eta)$ is given is Eq. (21). Similar to the bosonic case, we truncate the expression to the linear order in the gauge field

$$\hat J_s(x) = P(\hat\partial_1, \hat\partial_2)\bar\psi(x_1)\hat\gamma\psi(x_2)\phi(x_2) + iQ(\hat\partial_1, \hat\partial_2, \hat\partial_3)\bar\psi(x_1)\hat\gamma\psi(x_2)\hat A(x_3)\big|_{x_{1,2,3}\to x}$$
$$= \hat J_s^{(P)}(x) + \hat J_s^{(Q)}(x)\,, \tag{72}$$

where

$$Q(\xi, \eta, \chi) = \frac{P(\xi+\chi, \eta) - P(\xi, \eta+\chi)}{\chi}\,. \tag{73}$$

We then calculate its divergence

$$\hat K_{s-1} = \partial_\mu D_z^\mu \hat J_s\,. \tag{74}$$

The divergence of the $A$-independent part $\hat J_s^{(P)}$ is

$$\hat K_{s-1}^{(P)} = \left[M_1(\hat\partial_1, \hat\partial_2)\partial_1^2 + M_2(\hat\partial_1, \hat\partial_2)\partial_2^2\right]\bar\psi(x_1)\hat\gamma\psi(x_2)$$
$$+ \left[N_1(\hat\partial_1, \hat\partial_2)\partial_1^\lambda + N_2(\hat\partial_1, \hat\partial_2)\partial_2^\lambda\right]\bar\psi(x_1)\gamma_\lambda\psi(x_2)\,, \tag{75}$$

where the polynomials

$$M_1(\xi, \eta) = \frac{d}{2}P'_\xi + \frac{1}{2}(\xi-\eta)P'_{\xi\xi} + \eta P'_{\xi\eta}\,,$$
$$M_2(\xi, \eta) = \frac{d}{2}P'_\eta - \frac{1}{2}(\xi-\eta)P'_{\eta\eta} + \xi P'_{\xi\eta}\,,$$
$$N_1(\xi, \eta) = \frac{d-2}{2}P + \xi(P'_\eta - P'_\xi)\,,$$
$$N_2(\xi, \eta) = \frac{d-2}{2}P + \eta(P'_\xi - P'_\eta)\,. \tag{76}$$

We then substitute in the equations of motion of (69). The divergence of the $A$-dependent part $\hat{J}_s^{(Q)}$ can be calculated similarly. For this part, one can simply use the free equation of motion $\not{\partial}\psi = 0$. Combined together, the final result is gauge invariant, dependent only on the field strength $F_{\mu\nu} = \partial_\mu A_\nu - \partial_\nu A_\mu$.

$$
\begin{aligned}
\hat{K}_{s-1} = & \left[R_1(\hat{\partial}_1, \hat{\partial}_2, \hat{\partial}_3)\partial_1^\mu + R_2(\hat{\partial}_1, \hat{\partial}_2, \hat{\partial}_3)\partial_2^\mu + R_3(\hat{\partial}_1, \hat{\partial}_2, \hat{\partial}_3)\partial_3^\mu\right] i\bar{\psi}_1\hat{\gamma}F_{3\mu\nu}z^\nu\psi_2 \\
& + R_4(\hat{\partial}_1, \hat{\partial}_2, \hat{\partial}_3)\bar{\psi}_1 F_{3\mu\nu}z_\lambda\epsilon^{\mu\nu\lambda}\psi_2 + R_5(\hat{\partial}_1, \hat{\partial}_2, \hat{\partial}_3)i\bar{\psi}_1 F_{3\mu\nu}\gamma^\mu z^\nu\psi_2\big|_{x_{1,2,3}\to x}\,, \quad (77)
\end{aligned}
$$

where the polynomials

$$
\begin{aligned}
R_1(\xi, \eta, \chi) &= \frac{2}{\chi}M_1(\xi+\chi, \eta) - \frac{1}{\chi}\left(\frac{d}{2}Q - (\eta+\chi)Q'_\xi + \eta Q'_\eta + \chi Q'_\chi\right), \\
R_2(\xi, \eta, \chi) &= -\frac{2}{\chi}M_2(\xi, \eta+\chi) - \frac{1}{\chi}\left(\frac{d}{2}Q + \xi Q'_\xi - (\xi+\chi)Q'_\eta + \chi Q'_\chi\right), \\
R_3(\xi, \eta, \chi) &= \frac{1}{\chi}(M_1(\xi+\chi, \eta) - M_2(\xi, \eta+\chi)) - \frac{1}{\chi}\left(\frac{d}{2}Q + \xi Q'_\xi + \eta Q'_\eta - (\xi+\eta)Q'_\chi\right), \quad (78) \\
R_4(\xi, \eta, \chi) &= -\frac{1}{2}(M_1(\xi+\chi, \eta) + M_2(\xi, \eta+\chi)), \\
R_5(\xi, \eta, \chi) &= (M_1(\xi+\chi, \eta) - M_2(\xi, \eta+\chi)) + \frac{1}{\chi}(N_1(\xi+\chi, \eta) - N_2(\xi, \eta+\chi)) + \frac{\xi+\eta+\chi}{\chi}Q.
\end{aligned}
$$

Especially, for $s=1$ adjoint, $(\hat{K}_0)^i_{\ j} = 0$, and for $s=2$ singlet, $\hat{K}_1 = 6i\bar{\psi}_{k,1}\gamma^\mu\psi_2^k F_{\mu\nu,3}z^\nu = 0$ due to the equation of motion for $A_\mu$ in $3d$, which corresponds to the conservation of symmetry current and stress tensor.

In the adjoint sector, the correlation of the higher-spin operator $\hat{J}_s$ and the divergence $\hat{K}_{s-1}$ can be evaluated by direct contraction. Substitute the correlation functions into Eq. (26), we get the result for the anomalous dimension in $3d$

$$
\gamma_s^{\text{QED, ad}} = \frac{16}{N\pi^2}\left[\sum_{i=1}^s \frac{1}{i-1/2} - \frac{2}{3}\frac{11s^2-2}{4s^2-1}\right]. \quad (79)
$$

Note that this result is exactly the same as tricritical QED. Based on this, we conjecture that the anomalous dimension of the tricritical QED and fermionic $\text{QED}_3$ in the singlet sector should also be the same. We verify this coincidence up to $s=5$ by calculating leading order diagrams of the $\hat{K}_{s-1}$ correlators

$$
\langle\hat{K}_{s-1}(x)\hat{K}_{s-1}(0)\rangle = \quad \text{} \quad (80)
$$

## 4.2 QED$_3$-Gross-Neveu-Yukawa theory

In this section we further couple the fermions in QED to an auxiliary bosonic field through a Gross-Neveu-Yukawa interaction. The modified equation of motion for $\psi$ and $\bar{\psi}$ are, to linear order of $A$ and $\sigma$

$$
\begin{aligned}
\not{\partial}\psi &= i\not{A}\psi - \sigma\psi\,, \\
\partial_\mu\bar{\psi}\gamma^\mu &= -i\bar{\psi}\not{A} + \bar{\psi}\sigma\,, \\
\partial^2\psi &= \frac{i}{2}\gamma_\lambda\epsilon^{\mu\nu\lambda}F_{\mu\nu}\psi + i(\partial_\mu A^\mu)\psi + 2iA^\mu(\partial_\mu\psi) - (\not{\partial}\sigma)\psi\,, \\
\partial^2\bar{\psi} &= \frac{i}{2}\bar{\psi}F_{\mu\nu}\gamma_\lambda\epsilon^{\mu\nu\lambda} - i\bar{\psi}(\partial_\mu A^\mu) - 2i(\partial_\mu\bar{\psi})A^\mu + \bar{\psi}(\not{\partial}\sigma)\,. \quad (81)
\end{aligned}
$$

This modification results in an extra piece in the divergence

$$\hat{K}_{s-1} = \hat{K}_{s-1}^{(\text{QED})} + \hat{K}_{s-1}^{(\text{GNY})}, \tag{82}$$

where $\hat{K}_{s-1}^{(\text{QED})}$ is the divergence in QED given in the Eq. (77), and

$$\hat{K}_{s-1}^{(\text{GNY})} = R_6(\hat{\partial}_1, \hat{\partial}_2, \hat{\partial}_3)\bar{\psi}_1 \sigma_3 \psi_2 + R_7(\hat{\partial}_1, \hat{\partial}_2, \hat{\partial}_3)\bar{\psi}_1 (\partial_3^\mu \sigma_3) z^\nu \gamma^\lambda \epsilon_{\mu\nu\lambda} \psi_2, \tag{83}$$

the polynomials are

$$R_6(\xi, \eta, \chi) = \chi \left[ M_1(\xi + \chi, \eta) - M_2(\xi, \eta + \chi) \right] + N_1(\xi + \chi, \eta) - N_2(\xi, \eta + \chi),$$
$$R_7(\xi, \eta, \chi) = M_1(\xi + \chi, \eta) + M_2(\xi, \eta + \chi). \tag{84}$$

In the adjoint sector, the correlation of the higher-spin operator $\hat{J}_s$ is the same as in QED, and the correlation of the divergence $\hat{K}_{s-1}$ can be evaluated by direct contraction. Substitute the correlation functions into Eq. (26), we get the result for the anomalous dimension in 3$d$

$$\gamma_s^{(\text{QED−GNY,ad})} = \frac{16}{N\pi^2} \left( \sum_{i=1}^{s} \frac{1}{i - 1/2} - \frac{7s^2 - 2}{4s^2 - 1} \right). \tag{85}$$

We note that this result is the same as scalar QED. We also expect $\gamma_s$ in the singlet sector is the same as that of scalar QED.

# 5 SU($N$)$_1$ WZW CFT in $(2 + \epsilon)$

It is well known that starting from a Gaussian theory at the upper or lower critical dimensions, one can perform dimensional continuation to obtain and to calculate interacting CFTs perturbatively, which include: (1) $d = 4 - \epsilon$ dimensional Wilson-Fisher [45, 46], Gross-Neveu-Yukawa [47, 48], critical gauge theories with bosonic and/or fermionic matter [49–53]; (2) $d = 2 + \epsilon$ dimensional non-linear sigma models with no topoloigcal terms[5] [54]; (3) $d = 2 + \epsilon$ dimensional Gross-Neveu-Yukawa theory [55]. Conformal data such as scaling dimensions of operators can be written as a series expansion of $\epsilon$, and the series is known to be a divergent asymptotic series. One physical reason for the series expansion being divergent is that $d = 2, 4$ dimensions are the branch cut of the theory at which the Gaussian fixed point merges with the interacting fixed point. This naturally brings a question: can we perturbatively define and calculate an interacting CFT starting from a non-Gaussian (but solvable) theory? An ideal starting point is the 2$d$ CFT, in particular, the Ising CFT is believed to exist in $2 \leq d \leq 4$ dimensions [56, 57].

Similar to free theories, 2$d$ CFTs also have conserved higher-spin currents as a consequence of the Virasoro symmetry. However, it remains a mystery about how these conserved higher-spin currents are broken if the theory is continued to $d = 2 + \epsilon$ dimensions. The idea of conformal multiplet recombination does not naively apply here. Specifically, the divergence equation $\partial \cdot J_s = K_{s-1}$ requires a spin-$(s-1)$ operator $K_{s-1}$ with $\Delta = s + 1$. Such an operator, however, does not exist in a generic 2$d$ CFT's spectrum (viz. minimal models, WZW models) even if null operators are taken into account. On the other hand, there is no known way to get around the conformal multiplet recombination, as one can rigorously show that $\partial \cdot J_s$ has to be a primary operator of the global conformal symmetry [32]. In this section, we will

---

[5]These non-linear sigma models are believed to describe the same fixed point as Wilson-Fisher bosons with/without gauge fields.

provide a solution to this mystery for the SU$(N)_1$ WZW models, and it can be straightforwardly generalised to other WZW models.

The idea is to consider a dual description of the SU$(N)_1$ WZW CFT, namely a fermionic QED with $N$ Dirac fermions coupled to a U(1) gauge field. This is the only known description that can be generalised into $(2+\epsilon)$-dimensions. This QED$_2$ theory is also called the Schwinger model. It can be exactly solved using the bosonisation technique. One important feature of the exact solution is that the gauge field strength $F_{\mu\nu}$ as well as any operator proportional to it will decouple from the IR spectrum.[6] As we have explicitly shown in previous sections, in gauge theories higher-spin currents get broken by 'eating' the divergence operator $K_{s-1}$ which is proportional to $F_{\mu\nu}$. The absence of divergence operator can also be understood from the fact that there exist no operator with spin-$(s-1)$ and scaling dimension $\Delta = s+1$. So the decoupling of $F_{\mu\nu}$ and its composite operators will make higher-spin currents conserved, and also explains why the divergence operators are absent in the spectrum of SU$(N)_1$ WZW models. More importantly, these operators only decouple at $2d$, they will re-enter the spectrum at $d = 2+\epsilon$ dimensions making higher-spin currents slightly broken.

Concretely, the correlator of $F_{\mu\nu}$ can be written as

$$\langle F_{\mu\nu}(x)F_{\rho\sigma}(0)\rangle = C(\epsilon,N)\frac{I_{\mu\rho}I_{\nu\sigma} - I_{\mu\sigma}I_{\nu\rho}}{x^4}\,, \tag{86}$$

where $I_{\mu\nu}(x) = \delta_{\mu\nu} - 2\frac{x_\mu x_\nu}{x^2}$. $F_{\mu\nu}$ decouples at $2d$ corresponds to $C(\epsilon = 0, N) = 0$, and it is a non-perturbative statement. To gain a more quantitative understanding, we can consider the large-$N$ limit, where the correlator of $F_{\mu\nu}$ at arbitrary $2 \le d \le 4$ and up to the order of $1/N^2$ is,

$$C(\epsilon = d-2, N) = \frac{-\Gamma(d)\sin\frac{\pi d}{2}}{\pi\Gamma(\frac{d}{2})^2}\frac{1}{N} + \frac{\Gamma(d)\sin\frac{\pi d}{2}}{\pi\Gamma(\frac{d}{2})^2}\left[3\psi'\left(\frac{d}{2}\right) - \frac{\pi^2}{2} + \frac{4(d-1)}{d}\right]\frac{1}{N^2} + \mathcal{O}\left(\frac{1}{N^3}\right)\,, \tag{87}$$

where $\psi(x)$ is the digamma function. The $\mathcal{O}(1/N^2)$ correction is given by Ref. [58]. It is consistent with the fact that they will decouple at $2d$, namely $\langle F_{\mu\nu}(x)F_{\rho\sigma}(0)\rangle = 0 + \mathcal{O}(1/N^3)$. In the regime $1/N \ll \epsilon \ll 1$,

$$C(\epsilon, N) = 2\epsilon/N + 2\epsilon^2/N^2 + O(1/N^3)\,. \tag{88}$$

So in this regime we have

$$\gamma_s^{\text{QED,ad}} = \frac{\epsilon}{N}\sum_{i=1}^{s-1}\frac{1}{i} + \mathcal{O}(\epsilon^2) = \frac{\epsilon}{N}H_{s-1} + \mathcal{O}(\epsilon^2)\,, \tag{89}$$

where $H_{s-1}$ is the Harmonic number. It is tempting to conjecture that the same result holds for the finite $N$. The indication is that, the $\mathcal{O}(1/N)$ correlator is of order $\mathcal{O}(\epsilon)$, while $\mathcal{O}(1/N^2)$ correlator is of order $\mathcal{O}(\epsilon^2)$. It is possible that higher order $\mathcal{O}(1/N^k)$ $(k > 2)$ correlator is also of order $\mathcal{O}(\epsilon^2)$ (or higher). It will be great to have a non-perturbative proof to justify this conjecture.

---

[6]Intuitively, it can be understood from the fact that U(1) gauge field is always linearly confined in $2d$.

# 6 Conclusion and Discussion

In the previous sections, we have calculated the anomalous dimensions of various bosonic and fermionic QEDs. We find these results have similar logarithmic asymptotic behaviours in the large-$s$ limit

$$\gamma_s = \frac{\text{const.}}{N} \log s + \dots \qquad (s \to \infty), \tag{90}$$

which is different from non-gauge interacting CFTs (i.e. Wilson-Fisher, Gross-Neveu-Yukawa), $\gamma_s = \text{const.}/N + \cdots$. Results from light-cone bootstrap [15–17] provide an explanation for this difference. For any primary (scalar) operator $O$ in a unitary CFT, its twist family with scaling dimensions, $\Delta = 2\Delta_O + 2n + s + \mathcal{O}(1/s)$, will always exist in the CFT's operator spectrum.[7] The slightly broken higher-spin currents of the Wilson-Fisher theory is just a twist family of $\phi$, hence their $\gamma_s = 2\Delta_\phi - 1 + \mathcal{O}(1/s) = \text{const.}/N + \cdots$. In contrast, a gauge theory does not have $\phi$ in its spectrum (since they are not gauge invariant), so its $\gamma_s$ does not have to follow the behavior $\gamma_s = \text{const.}/N + \cdots$.

However, we would like to emphasise that our results do not apply to the real large-$s$ limit. Since we perform the large-$N$ expansion in the first place, the results apply only to the case where $N$ is still the leading parameter compared to $s$.[8] To compare gauge theories and non-gauge theories in the large-$s$ limit, we need to extend our results from the region where $N$ is the leading scale to the region where $s$ is the leading scale. This region is not accessible by our large-$N$ expansion. There are several possibilities how $\gamma_s$ would extend from the large-$s$ region to the large-$N$ region. One possibility is the logarithmic divergence continues;[9] the other possibility is that $\gamma_s$ converges to a finite limit, for example, one possibility is

$$\gamma_s \sim \gamma_\infty(1 - s^{-a/N}), \tag{91}$$

where $\gamma_\infty$ is an $\mathcal{O}(1/N^0)$ constant.

One question to ask is whether the slightly broken spin-$s$ current is still the spin-$s$ operator with the minimal twist in gauge theories.[10] One candidate for the minimal twist is the twist family of the SU($N$) conserved current, which has $\lim_{s\to\infty} \tau'_s = 1$. We can compare this with $\gamma_s$ of the slightly broken higher-spin currents. There are several possible scenarios: (1) Either $\gamma_s$ diverges logarithmically, or converges to a limit $\gamma_\infty > 1$, then the minimal twist is $\tau'_s$; (2) $\gamma_s$ converges to a limit $\gamma_\infty < 1$, hence the slightly broken higher-spin currents are the minimal twist. Thanks to the lightcone bootstrap result [17], we know that $\gamma_s$ as a function of $s$ must be convex. Thus, in either scenario, the minimal twist in the large-$s$ limit $\tau_{\infty,\text{min}} = \lim_{s\to\infty} \tau_{s,\text{min}}$ converges slower than $1/N$ in the large $N$ limit, namely

$$\lim_{N\to\infty} \tau_{\infty,\text{min}} N = \lim_{N\to\infty} \left( \lim_{s\to\infty} \tau_{s,\text{min}} N \right) = \infty. \tag{92}$$

This is a crutial difference between gauge theories and interacting theories without gauge theories, as the latter has $\lim_{N\to\infty} \tau_{\infty,\text{min}} N = \mathcal{O}(1/N^0)$.

We have also discussed SU($N$)$_1$ WZW CFT at $2 + \epsilon$ dimensions using its dual QED description. Specifically, we argue that the conservation of higher-spin currents of SU($N$)$_1$ WZW CFT at $2d$ can be understood as a consequence of the gauge field strength being decoupled in the IR. At $d = 2 + \epsilon$ dimensions the gauge field strength re-enters the operator spectrum, and

---

[7]Schematically, we can write these operators as $O\partial^s\Box^n O$.

[8]We conjecture that the limit could be written as $1 \ll \log s \ll N$, as in this case, the correction to the scaling dimension $\gamma_s \ll 1$ is small.

[9]We would like to note that for the Wilson-Fisher theory, the large-$N$ expansion result is identical to the light-cone bootstrap result, although the former applies to the region where $N$ is the leading scale while the latter is valid for the region where $s$ is the leading scale.

[10]The twist is defined as $\tau_s = \Delta - s - 1$.

breaks higher-spin currents through conformal multiplet recombination. It is worth emphasising that different from the conformal multiplet recombination in other previously known cases, here the divergence operators (i.e. the operators being 'eaten') are not in the original operator spectrum of the $2d$ theory. This new type of conformal multiplet recombination may be present in the dimensional continuation of many other $2d$ CFTs, in particular the Ising CFT. We will leave this to the future study.

At last, we would like to comment on the possible physical or experimental correspondence of the anomalous dimensions of slightly broken higher-spin currents. An intriguing possibility is that they may be related to the non-equilibrium properties of CFTs. The intuition behind this is that in the presence of higher-spin symmetry, the system is known to be integrable. The breaking of their conservation will spoil integrability, hence yielding non-equilibrium phenomena like thermalisation or scrambling.

## Acknowledgements

We would like to thank Jaume Gomis, Ning Su and Liujun Zou for discussions.

**Funding information**   Z. Z. acknowledges supports from the Natural Sciences and Engineering Research Council of Canada (NSERC) through Discovery Grants. Research at Perimeter Institute is supported in part by the Government of Canada through the Department of Innovation, Science and Industry Canada and by the Province of Ontario through the Ministry of Colleges and Universities.

## A   Calculation details for the singlet sector

In this appendix, we use tricritical QED as an example to demonstrate how we calculate the singlet sector anomalous dimensions. In tricritical QED, the singlet sector higher-spin operator $\hat{J}_s$ and its divergence $\hat{K}_{s-1}$ are

$$
\hat{J}_s = P(\hat{\partial}_1, \hat{\partial}_2)\bar{\phi}_{j,1}\phi_2^i + iQ(\hat{\partial}_1, \hat{\partial}_2, \hat{\partial}_3)i\bar{\phi}_{k,1}\phi_2^k\hat{A}_3 \,,
$$
$$
\hat{K}_{s-1} = \mathscr{K}_{(A)}^{\mu\nu}(\partial_1, \partial_2, \partial_3)i\bar{\phi}_{k,1}\phi_2^k F_{\mu\nu,3} \,. \tag{A.1}
$$

Note that the correlators of $\hat{J}_s$ and $\hat{K}_{s-1}$ are no longer only contractions. To the leading order of $1/N$, we have to consider other possible Feynman diagrams. For the correlation of $\hat{J}_s$, we can still omit the $\hat{A}$-dependent piece, and

$$
\left\langle \hat{J}_s(x)\hat{J}_s(0) \right\rangle = P(\hat{\partial}_1, \hat{\partial}_2)P(\hat{\partial}_{1'}, \hat{\partial}_{2'})\left\langle (\bar{\phi}_{1,k}\phi_2^k)(\bar{\phi}'_{1,l}\phi_2'^l) \right\rangle_\infty \Big|_{x_{1,2}\to x, x'_{1,2}\to 0}
$$

$$
= \text{(a)} + \text{(b)} \,, \tag{A.2}
$$

where a ring dot denotes that differential operators are acted on this point. The diagram (b) vanishes identically for $s \geq 2$ due to the orthogonality between different spin currents $\langle \hat{J}_s(x)\hat{J}_1(0)\rangle = 0$. For $s = 1$, the spin-1 singlet current is the gauge current that is removed from the operator spectrum.

For the correlation of $\hat{K}_s$,

$$\langle\hat{K}_{s-1}(x)\hat{K}_{s-1}(0)\rangle = \mathcal{K}_{(A)}^{\mu\nu}(\partial_1,\partial_2,\partial_3)\mathcal{K}_{(A)}^{\rho\sigma}(\partial_{1'},\partial_{2'},\partial_{3'})\langle(\bar{\phi}_{1,k}\phi_2^k F_3^{\mu\nu})(\bar{\phi}_{1',l}\phi_{2'}^l F_{3'}^{\rho\sigma})\rangle\Big|_{x_\alpha\to x, x_\alpha'\to 0}$$

$$= \underset{(a)}{\text{diagram}} + \underset{(b)}{\text{diagram}} + \underset{(c)}{\text{diagram}}. \tag{A.3}$$

Here diagram (a) corresponds to direct contraction. Diagram (c) vanishes identically due to the orthogonality between slightly broken higher-spin currents and the field strength, namely $\langle\hat{J}_s(x)F^{\mu\nu}(0)\rangle = 0$. Taking into account diagram (b) is equivalent to making use of the equation of motion for $A_\mu$ to the zeroth order of $A$

$$0 = i(\partial_{1\mu} - \partial_{2\mu})\bar{\phi}^k(x_1)\phi_k(x_2)\Big|_{x_{1,2}\to x} + \mathcal{O}(A), \tag{A.4}$$

which effectively remove the pieces proportional to $\hat{J}_1$. To show this, we rewrite the divergence in terms of the slightly broken higher-spin currents, the field strength and their descendents.

$$\hat{K}_{s-1} = \sum_{l=0}^{s-2}[J_l][F], \tag{A.5}$$

where $[\dots]$ denotes the conformal family of the operator, and in $J_l$ we need only to keep the $\hat{A}$-independent piece. More explicitly, one can write the descendents in the form

$$[J_l][F] = \sum_{m=0}^{s-1-l}\Big(a_{slm}\hat{\partial}^{s-l-m-1}D_z^\mu\hat{J}_l\hat{\partial}^m F_{\mu\nu}z^\nu$$
$$+b_{slm}\hat{\partial}^{s-l-m-2}\partial^\mu\hat{J}_l\hat{\partial}^m F_{\mu\nu}z^\nu + c_{slm}\hat{\partial}^{s-l-m-2}\hat{J}_l\hat{\partial}^m\partial^\mu F_{\mu\nu}\Big). \tag{A.6}$$

The coefficeents $a_{slm}, b_{slm}, c_{slm}$ can be determined by writing out $\hat{J}_l$ explicitly and compare the coeffecients with Eq. (44). In particular, due to the differential equation of $P(\xi,\eta)$, the coeffecient with $l=1$ can be determined explicitly

$$a_{m1} = \frac{2}{m!(s-2-m)!(s-m)!}\partial_\chi^m D_P^{s-2-m}(R_1-R_2)\big|_{\chi=0},$$

$$\frac{d-2}{2}a_{m1} + b_{m1} = \frac{1}{m!(s-2-m)!(s-m)!}\partial_\chi^m D_P^{s-2-m}(\xi R_1+\eta R_2)\big|_{\xi=1,\eta=-1,\chi=0}, \tag{A.7}$$

$$c_{m1} = \frac{1}{m!(s-3-m)!(s-1-m)!}\partial_\chi^m D_P^{s-3-m}k_3\big|_{\xi=1,\eta=-1,\chi=0},$$

where the differential operator

$$D_P = \frac{d-2}{2}(\partial_\xi+\partial_\eta)+\xi\partial_\xi^2+\eta\partial_\eta^2. \tag{A.8}$$

We then evaluate the diagrams in Eq. (A.3) using this form of $\hat{K}_{s-1}$.

$$\text{(a)} = \left\langle\left(\sum_{l=0}^{s-2}[J_l][F]\right)(x)\left(\sum_{l'=0}^{s-2}[J_{l'}][F]\right)(0)\right\rangle_{\text{ct.}} = \sum_{l=0}^{s-2}\langle([J_l][F])(x)([J_l][F])(0)\rangle_{\text{ct.}}, \tag{A.9}$$

where 'ct.' means direct contraction. For diagram (b), we note that the bubbles are non-zero only when $l=1$ due to the orthogonality between slightly broken higher-spin currents with different spin

$$\text{diagram} = \left\langle\left(\sum_{l=0}^{s-2}[J_l]\right)J_1^\mu\right\rangle_{\text{ct.}} = \sum_{l=0}^{s-2}\langle[J_l]J_1^\mu\rangle_{\text{ct.}}\delta_{l,1} = \langle[J_1]J_1^\mu\rangle_{\text{ct.}}. \tag{A.10}$$

With this result, the chain integral [59] in diagram (b) is evaluated to be

$$(b) = -\langle ([J_1][F])(x)([J_1][F])(0)\rangle_{\text{ct.}}.$$  (A.11)

Therefore, the correlation of $\hat{K}_{s-1}$ in large-$N$ limit can be written effectively as the contraction of $\hat{\bar{K}}_{s-1}$ in which the terms proportional to $J_1$ removed from $\hat{K}_s$.

$$\langle \hat{K}_{s-1}(x)\hat{K}_{s-1}(0)\rangle_\infty = \langle \hat{\bar{K}}_{s-1}(x)\hat{\bar{K}}_{s-1}(0)\rangle_{\text{ct.}}, \qquad \hat{\bar{K}}_{s-1} = \hat{K}_{s-1} - [J_1][F].$$  (A.12)

# B  Bosonic QEDs with Chern-Simons term

In this section we add an additional Chern-Simons term to the bosonic theories,

$$\mathcal{L}_{\text{CS}} = \frac{ik}{4\pi}\epsilon^{\mu\nu\lambda}A_\mu\partial_\nu A_\lambda.$$  (B.1)

We consider the limit of large CS-level $k$ and finite $\lambda = k/N$. This results in an extra factor in front of the photon propagator [25]

$$\langle A^\mu(x)A^\nu(0)\rangle_{\text{CS}} = \frac{1}{1+\lambda^2}\langle A^\mu(x)A^\nu(0)\rangle_\infty$$
$$= \frac{1}{1+\lambda^2}\frac{1}{8\pi^2 N}\frac{(d-2-\zeta)\delta^{\mu\nu} + 2\zeta\frac{x^\mu x^\nu}{x^2}}{x^2},$$  (B.2)

where $\langle\ldots\rangle_\infty$ are correlators evaluated with the Feynman rules without Chern-Simons term. and the equation of motion for the gauge field $A$ is modified to

$$\epsilon^{\lambda\mu\nu}F_{\mu\nu} = \frac{4\pi i}{k}(\partial_1^\lambda - \partial_2^\lambda)\bar{\phi}_k(x_1)\phi^k(x_2)|_{x_{1,2}\to x} = \frac{4\pi}{k}J_1^\lambda.$$  (B.3)

In the adjoint sector, we need only take the $1/(1+\lambda^2)$ factor in Eq. (B.2) into account when calculating the correlation of the piece proportional to $F_{\mu\nu}$ in $\hat{K}_{s-1}$, and the result is modified to

$$\gamma_s^{\text{tr.QED+CS,ad}} = \frac{16}{N\pi^2}\frac{1}{1+\lambda^2}\left[\sum_{i=1}^s\frac{1}{i-1/2} - \frac{2(11s^2-2)}{3(4s^2-1)}\right]$$  (B.4)

in tricritical QED, and

$$\gamma_s^{\text{sc.QED+CS,ad}} = \frac{16}{N\pi^2}\left[\frac{1}{1+\lambda^2}\sum_{i=1}^s\frac{1}{i-1/2} + \frac{s^2-1}{3(4s^2-1)} - \frac{1}{1+\lambda^2}\frac{2(11s^2-2)}{3(4s^2-1)}\right]$$  (B.5)

in scalar QED.

In the singlet sector, we also need to take into account the equation of motion for the gauge field $A$ (B.3) which relates the field strength to the gauge current. Substituting $F$ with $J_1$, we get

$$\hat{K}_{s-1} = \sum_{l=0}^{s-2}[J_l][J_1].$$  (B.6)

Its correlation function is

$$\langle \hat{K}_{s-1}\hat{K}_{s-1}\rangle_{\text{CS}} = \frac{1}{1+\lambda^2}\sum_{l\neq 1}\left\langle (\overbrace{[J_l]\overbrace{[J_1]})(x)([J_l]}[J_1])(0)\right\rangle_\infty$$
$$+ \frac{1}{(1+\lambda^2)^2}\left[\left\langle (\overbrace{[J_1]\overbrace{[J_1]})(x)([J_1]}[J_1])(0)\right\rangle_\infty + \left\langle (\overbrace{[J_1][J_1]})(x)(\overbrace{[J_1][J_1]})(0)\right\rangle_\infty\right],$$  (B.7)

where the contraction $\overparen{J_m J_n}$ is a shorthand notation for $(P_m(\hat{\partial}_1, \hat{\partial}_2)\overline{\overparen{\phi_1 \phi_2}})(P_n(\hat{\partial}_{1'}, \hat{\partial}_{2'})\overline{\overparen{\phi_{1'} \phi_{2'}}})$. Plugging it into Eq. (26), we get the result for the anomalous dimension

$$\gamma_s^{\text{tr.QED+CS,sg}} = \frac{16}{N\pi^2}\frac{1}{1+\lambda^2}\left[\sum_{i=1}^s \frac{1}{i-1/2}\right.$$
$$\left. -\begin{cases} \dfrac{2(11s^2-2)}{3(4s^2-1)}, & s \text{ odd} \\ \dfrac{2(11s^4-s^2+8)}{3(4s^4-5s^2+1)} + \dfrac{1}{1+\lambda^2}\dfrac{2(s-2)(s-1)}{(s+1)(4s^2-1)}, & s \text{ even} \end{cases}\right] \quad \text{(B.8)}$$

in tricritical QED, and

$$\gamma_s^{\text{sc.QED+CS,sg}} = \frac{16}{N\pi^2}\left[\frac{1}{1+\lambda^2}\sum_{i=1}^s \frac{1}{i-1/2}\right.$$
$$\left. +\begin{cases} \dfrac{s^2-1}{3(4s^2-1)} - \dfrac{1}{1+\lambda^2}\dfrac{2(11s^3+3s^2-2s-3)}{3s(4s^2-1)}, & s \text{ odd} \\ \dfrac{s-2}{6(2s-1)} - \dfrac{1}{1+\lambda^2}\dfrac{2(11s^4-s^2+8)}{3(4s^4-5s^2+1)} - \dfrac{1}{(1+\lambda^2)^2}\dfrac{2(s-2)(s-1)}{(s+1)(4s^2-1)}, & s \text{ even} \end{cases}\right] \quad \text{(B.9)}$$

in scalar QED.

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
