# Peer review of "Slightly broken higher-spin current in bosonic and fermionic QED in the large-$N$ limit"

_SciPost Physics, doi:SciPost Phys. 15, 072 (2023)_

## Round 1 · Referee Report · Anonymous (Referee 1) · 2023-4-14

Strengths

1- Detailed large-N calculation of anomalous scaling dimension of higher spin current with different representations in various QED3 models with bosonic and fermionic matter fields. 2- The authors use the duality between $SU(N)_1$ and fermionic gauge theory to resolve the puzzle of slightly broken higher-spin currents in the WZW model.

Weaknesses

1- Several typos in the manuscript, list a few: (2.1) and (2.2) the sub- and superscripts are disorganized. (2.15) variables do not match. Also typo in (3.4). 2- The calculation in this work is akin to the previous calculation of non-Abelian Chern-Simons theory with vector matter, the current results could be compared with previous ones. There are also dualities between these theories as in arXiv:1812.01544, the anomalous scaling dimension of these higher spin currents could be compared. 3- In the introduction, the authors claim to calculate the anomalous scaling dimension of higher spin current of $SU(N)_1$ WZW at 2+$\epsilon$ dimension. This is misleading since the calculation in Sec. 5 is of dual fermionic QED theory. More evidence needs to provide for the duality to hold in the $2+\epsilon$ dimension.

Report

The authors investigated the slightly broken higher-spin current in QED3 theories with the bosonic and fermionic matter. The authors present a comprehensive calculation of the anomalous scaling dimensions of these operators.
These models are relevant to the quantum critical points in the condensed matter, the analysis of higher-spin current could help bootstrap these models.
These models also form a duality web which is important for understanding the relationship between various 3d QFTs, with the data of higher-spin current, the duality web can be further constrained.
Therefore, if the points raised in "Weakness" can be addressed reasonably, the manuscript should be published.

Requested changes

Please see the "Weakness".

  • validity: high
  • significance: ok
  • originality: ok
  • clarity: ok
  • formatting: good
  • grammar: good

Author:  Zheng Zhou  on 2023-06-01  [id 3702]

(in reply to Report 1 on 2023-04-14)

Strengths

Detailed large-$N$ calculation of anomalous scaling dimension of higher spin current with different representations in various QED$_3$ models with bosonic and fermionic matter fields.

The authors use the duality between $\textrm{SU}(N)_1$ and fermionic gauge theory to resolve the puzzle of slightly broken higher-spin currents in the WZW model.

We thank the Referee for reviewing our paper and the positive evaluation.

Weaknesses

Several typos in the manuscript, list a few: (2.1) and (2.2) the sub- and superscripts are disorganized. (2.15) variables do not match. Also typo in (3.4).

We thank the Referee for the careful reading of our paper. We have fixed the mistakes in Eqs. (2.15) and (3.4). We have also modified Eqs. (2.1) and (2.2) so that the subscripts and superscripts align with [S. Giombi and V. Kirilin, J. High Energy Phys. 11, 068 (2016)].

The calculation in this work is akin to the previous calculation of non-Abelian Chern-Simons theory with vector matter, the current results could be compared with previous ones. There are also dualities between these theories as in arXiv:1812.01544, the anomalous scaling dimension of these higher spin currents could be compared.

The calculation for the non-Abelian Chern-Simons theory [S. Giombi, V. Gurucharan, V. Kirilin et al., J. High Energy Phys. 01, 058 (2017)] deals with a single fermion in $\mathrm{U}(N)$ foundamental representation coupled to a $\mathrm{U}(N)$ gauge field, while our calculation deals with $N$ fermions coupled to a $\mathrm{U}(1)$ gauge field. As the gauge fields and the limits taken are different, the two results cannot directly compare with each other. The dualities mentioned in arXiv:1812.01544 applies only to the case where the number of flavours $N_f=2$. As our manuscript deals with the large-$N_f$ limit, $N_f=2$ is too small to compare with our results.

In the introduction, the authors claim to calculate the anomalous scaling dimension of higher spin current of $\mathrm{SU}(N)_1$ WZW at $2+\epsilon$ dimension. This is misleading since the calculation in Sec. 5 is of dual fermionic QED theory. More evidence needs to provide for the duality to hold in the $2+\epsilon$ dimension.

We thank the Referee for pointing this out. $\mathrm{U}(1)$ gauge theory with $N$ Dirac fermions is the only known description of $\mathrm{SU}(N)_1$ WZW that can be to generalised into $(2+\epsilon)$-dimensions. We have also noted that in the revised manuscript by adding a sentence 'This is the only known description that can be generalised into $(2+\epsilon)$-dimensions. ' in Section 5, Paragraph 3.

Report

The authors investigated the slightly broken higher-spin current in QED$_3$ theories with the bosonic and fermionic matter. The authors present a comprehensive calculation of the anomalous scaling dimensions of these operators. These models are relevant to the quantum critical points in the condensed matter, the analysis of higher-spin current could help bootstrap these models. These models also form a duality web which is important for understanding the relationship between various 3d QFTs, with the data of higher-spin current, the duality web can be further constrained. Therefore, if the points raised in "Weakness" can be addressed reasonably, the manuscript should be published.

We thank the Referee for the summary and the positive evaluation of our paper. We hope that with the reply above and the changes made, the Referee could agree that the manuscript is suitable for further consideration for SciPost Phys.

---

## Round 1 · Referee Report · Anonymous (Referee 2) · 2023-4-17

Strengths

  1. Nice idea of a concrete computation.

Weaknesses

  1. The limit is not very precisely defined in the paper.
  2. The error terms are not well controlled in various approximation.
  3. It's not entirely clear what the overall punchline of the paper is.

Report

1.The calculation of the anomalous dimension is done for a fixed s in N\to\infty limit. It has been claimed that the result is valid for 1<< \log(s) <<N regime. When N\to\infty limit is taken, there is a leading 1/N term and there are error terms. The error terms can potentially and does depend on s. In the language of real analysis, the error is not uniform in s. To rephrase, even if for every fixed s, we can neglect the higher order terms in 1/N; in the large s limit, the error suppressed in 1/N can get enhanced by some function of s, the "non-uniformity" of error in s in the large N limit can bite back. The authors hope that the errors will be still be suppressed in the regime 1<< \log(s) <<N i.e one should take the limit N\to\infty, s\to\infty such that \frac{log(s)}{N} \to 0 limit. The rigorous way to show this is to estimate the s-dependent coefficient of 1/N^{k} terms in the expression for \gamma_s and show that in the limit \frac{log(s)}{N} \to 0 , they are indeed subleading compared to the main leading term. Why should this be true ? Or is it just an expectation ? I do understand that in physics, it is sometimes very hard to be rigorous but it would be nice to have the limiting procedure explained carefully and at least mention about the assumption under which one can prove/expect such results.

  1. The authors seem to suggest an explanation for why they see different behavior in gauge theory compared to the non-gauged ones using the lightcone bootstrap result in the first paragraph of the conclusion. Immediately after it , they discuss how the regime considered in the paper involves N as the largest parameter while in lightcone bootstrap spin is the largest parameter, so technically speaking lightcone bootstrap result does not seem to apply to the case considered by the authors. What is the correct explanation ?

Requested changes

  1. It would be nice to have the result stated and the limit defined precisely. The result reads: \lim \frac{N \gamma_s }{\log(s)} =constant What is the precise definition of limit ? is it just N\to \infty for any s uniformly ? But that contradicts the statement made in the paper regarding the regime of validity N>>log(s)>>1 . Thus I am led to believe that it is some sort of simultaneous limit, is it s\to\infty, N\to\infty such that \frac{\log(s)}{N} \to 0 ? Please see the point raised in the report.

  2. Clarify the point 2, raised in the report.

  3. Lightcone bootstrap has been rigorously studied and some of the basic statements has been proven as theorems very recently in https://arxiv.org/abs/2212.04893 . I think it would be helpful for the readers to mention this paper along with the original seminar papers [15] and [16].

  4. It would be helpful for the readers to put in bit more detailed explanation in section 5 regarding the absence of divergence operator in 2D and how it reappears in 2+\epsilon . It would be nice to clarify the limits here, both 1/\epsilon and N is large parameter here, is there any particular simultaneous limit being taken ?

  • validity: good
  • significance: good
  • originality: good
  • clarity: good
  • formatting: excellent
  • grammar: good

Author:  Zheng Zhou  on 2023-06-01  [id 3701]

(in reply to Report 2 on 2023-04-17)

Strength

Nice idea of a concrete computation.

We thank the Referee for reviewing our paper and the positive evaluation.

Weakness

The limit is not very precisely defined in the paper.

The error terms are not well controlled in various approximation.

We thank the Referee for raising these points. We will give a detailed response in the following.

It's not entirely clear what the overall punchline of the paper is.

The main points of our paper are,

  • To calculate the anomalous dimensions of the slightly broken higher-spin currents in various bosonic and fermionic QEDs in 3d, which will be valuable for numerically bootstrapping these 3d CFTs.
  • To investigate the difference between gauge theories and non-gauge theories from the perspective of lowest-lying spin-$s$ primary operator.
  • To understand the mechanism of multiplet recombination of 2d CFTs at $(2+\epsilon)$-dimensions.

Report

The calculation of the anomalous dimension is done for a fixed $s$ in $N\to\infty$ limit. It has been claimed that the result is valid for $1\ll\log(s)\ll N$ regime. When $N\to\infty$ limit is taken, there is a leading $1/N$ term and there are error terms. The error terms can potentially and does depend on $s$. In the language of real analysis, the error is not uniform in $s$. To rephrase, even if for every fixed $s$, we can neglect the higher order terms in $1/N$; in the large $s$ limit, the error suppressed in $1/N$ can get enhanced by some function of $s$, the "non-uniformity" of error in $s$ in the large $N$ limit can bite back. The authors hope that the errors will be still be suppressed in the regime $1\ll\log(s)\ll N$ i.e one should take the limit $N\to\infty$, $s\to\infty$ such that $\frac{\log(s)}{N} \to 0$ limit. The rigorous way to show this is to estimate the $s$-dependent coefficient of $1/N^{k}$ terms in the expression for $\gamma_s$ and show that in the limit $\frac{\log(s)}{N} \to 0$, they are indeed subleading compared to the main leading term. Why should this be true ? Or is it just an expectation ? I do understand that in physics, it is sometimes very hard to be rigorous but it would be nice to have the limiting procedure explained carefully and at least mention about the assumption under which one can prove/expect such results.

We expect our large-$N$ expansion to be valid only in the region where $N$ is the leading scale compared with $s$, i.e., we first take the limit of $N\to\infty$ while keeping $s$ finite, and then send $s$ to infinity. The only intuition why we conjecture that this limit could be written as $N\gg\log s$ is that in this case the correction to the scaling dimension $\gamma_s\sim\log s/N\ll 1$. However, it is beyond our scrope to give a rigorous mathematical description. In the revised manuscript, we have replaced the expression $1\ll\log s\ll N$ with the description 'where $N$ is the leading scale compared with $s$' and added a footnote to clarify the reason why we conjecture this limit could be written as $1\ll\log s\ll N$.

The authors seem to suggest an explanation for why they see different behavior in gauge theory compared to the non-gauged ones using the lightcone bootstrap result in the first paragraph of the conclusion. Immediately after it, they discuss how the regime considered in the paper involves $N$ as the largest parameter while in lightcone bootstrap spin is the largest parameter, so technically speaking lightcone bootstrap result does not seem to apply to the case considered by the authors. What is the correct explanation ?

We thank the Referee for raising this ambiguity. The region we want to compare is the large-spin limit where $s$ is the leading scale compared with $N$. This is the region where the lightcone bootstrap result for non-gauge theories applies. However, our large-$N$ calculation can only access the region where $N$ is the leading scale compared with $s$. Therefore, we discuss multiple scenarios how $\gamma_s$ will extend to the large-$s$ region and found that under all the possible scenarios $\lim_{N\to\infty}(\lim_{s\to\infty}\tau_{s,\textrm{min}}N)=\infty$ under the assumption that $\tau_{s,\mathrm{min}}$ is convex with respect to $s$, which is different from the non-gauge theories. To summarise, we compare the large-spin limit and discuss the extension of our large-$N$ result to that region where $s$ is the leading scale.

Requested changes

It would be nice to have the result stated and the limit defined precisely. The result reads : $\lim \frac{N \gamma_s }{\log(s)} =\mathrm{constant}$. What is the precise definition of limit ? is it just $N\to \infty$ for any $s$ uniformly ? But that contradicts the statement made in the paper regarding the regime of validity $N\gg\log(s)\gg 1$. Thus I am led to believe that it is some sort of simultaneous limit, is it $s\to\infty$, $N\to\infty$ such that $\frac{\log(s)}{N} \to 0$ ? Please see the point raised in the report.

In the manuscript, we have made clear that the large-$N$ expansion involving taking the limit $N\to\infty$ while keeping $s$ finite, and the asymptotic behaviour applies to the limit of large-$s$ while keeping $N$ still the dominant scale compared with $s$. We have removed the inaccurate expression $1\ll\log s\ll N$ and replace it with 'where $N$ is the leading scale compared with $s$' to avoid unnecessary inaccurate claims.

Clarify the point 2, raised in the report.

In Section 6, Paragraph 2, we added a discussion to calrify the limits we consider : 'However, we would like to emphasise that our results do not apply to the real large-$s$ limit. It is because we perform the large-$N$ expansion in the first place, the results apply only to the case where $N$ is still the leading parameter compared to $s$. To compare gauge theories and non-gauge theories in the large-$s$ limit, we need to extend our results from the region where $N$ is the leading scale to the region where $s$ is the leading scale. This region is not accessible by our large-$N$ expansion. There are several possibilities how $\gamma_s$ would extend from the large-$s$ region to the large-$N$ region...' We also note in a footnote that for Wilson-Fisher theory, the large-$N$ expansion result is identical to the light-cone bootstrap result, although the former applies to the region where $N$ is the leading scale while the latter is valid for the region where $s$ is the leading scale.

Lightcone bootstrap has been rigorously studied and some of the basic statements has been proven as theorems very recently in https://arxiv.org/abs/2212.04893. I think it would be helpful for the readers to mention this paper along with the original seminal papers [15] and [16].

We thank the Referee for proposing the relevant reference. We have cited this paper along with [15] and [16].

It would be helpful for the readers to put in bit more detailed explanation in Section 5 regarding the absence of divergence operator in 2D and how it reappears in $2+\epsilon$. It would be nice to clarify the limits here, both $1/\epsilon$ and $N$ is large parameter here, is there any particular simultaneous limit being taken ?

The absence of divergence operator in 2d is a non-perturbative statement. These divergence operators are built from the field strength $F_{\mu\nu}$, which always decouple from the theory in 2d. The decoupling of $F_{\mu\nu}$ can be seen from the exact solution of QED$_2$, i.e. Schwinger model, and intuitively it can be understood from the fact that $\mathrm{U}(1)$ gauge field is always linearly confined in 2d.

In $(2+\epsilon)$d the $U(1)$ gauge field will reappear in the spectrum (so are the divergence operators), but it is unclear how will it appear. Specifically, we want to consider the correlation function of $F_{\mu\nu}$

$$\langle F_{\mu\nu}(x)F_{\rho\sigma}(0)\rangle=C(\epsilon,N)\frac{I_{\mu\rho}I_{\nu\sigma}-I_{\mu\sigma}I_{\nu\rho}}{x^4},$$
The multiplet recombination tells us that at the leading order $\gamma_s = \frac{C(\epsilon, N)}{2} H_{s-1}$, but we are not sure about the precise form of $C(\epsilon, N)$ except $C(\epsilon=0, N)=0$. Next we consider the large-N limit (i.e. $N\gg 1/\epsilon\gg 1$), where one finds
$$C(\epsilon, N) = 2\epsilon/N + 2\epsilon^2/N^2 + O(1/N^3),$$
so in this limit we have $\gamma_s=\frac{\epsilon}{N} H_{s-1} +O(\epsilon^2)$, and we conjecture it may also hold for a finite $N$.

We have revised the main text to clarify this issue.

---

## Round 2 · Author Response

We are thankful to both Referees for carefully reviewing the paper and providing useful comments and suggestions, which helped us to improve the manuscript.

We appreciate the positive evaluation of both Referees. We also thank their critical comments that are important to understanding the problems in concern. These points are addressed in the attached reply. Thanks to the comments of the Referees, we have made improvements to the paper. The revisions made are listed below.

We give a point-by-point response to the comments of all Referees. We hope that with the changes made, the current manuscript will be considered suitable for further consideration in SciPost Physics.

---

## Round 2 · List of Changes

• We have added discussion on the simultaneous limits that we take for $s$ and $N$ in Section 6;

  • We have added discussion to clarify the absence of divergence operator in 2d and its reappearance in $(2+\epsilon)$d in Section 5;

  • We have added relevant references for lightcone bootstrap mentioned by Referee 1;

  • We have corrected several typos mentioned by Referee 2;

  • We have added discussion to clarify the generalisation of $\mathrm{SU}(N)_1$ WZW from 2d to $(2+\epsilon)$d in Section 5.

---

## Editorial Decision

published